# LPS-Induced Inhibition of miR-143 Expression in Brown Adipocytes Promotes Thermogenesis and Fever

**DOI:** 10.3390/ijms232213805

**Published:** 2022-11-09

**Authors:** Jie Liu, Dewei Zeng, Junyi Luo, Huan Wang, Jiali Xiong, Xingping Chen, Ting Chen, Jiajie Sun, Qianyun Xi, Yongliang Zhang

**Affiliations:** 1Guangdong Provincial Key Laboratory of Animal Nutrition Control, College of Animal Science, South China Agricultural University, Guangzhou 510642, China; 2Jiangxi Province Key Laboratory of Animal Nutrition, College of Animal Science and Technology, Jiangxi Agricultural University, Nanchang 330045, China

**Keywords:** LPS, BAT, thermogenesis, fever, miR-143, UCP1, AC9

## Abstract

Fever is an important part of inflammatory response to infection. Although brown adipose tissue (BAT) thermogenesis is known to be potently influenced by systemic inflammation, the role of BAT during infection-induced fever remains largely unknown. Here, we injected mice with a low dose of LPS and found that low-dose LPS can directly induce thermogenesis of brown adipocytes. It is known that miR-143 is highly expressed in the BAT, and miR-143 knockout mice exhibited stronger thermogenesis under cold exposure. Interestingly, miR-143 was negatively correlated with an LPS-induced increase of *TNFα* and *IL-6* mRNA levels, and the *IL-6* pathway may mediate the inhibition of miR-143 expression. Moreover, miR-143 is down-regulated by LPS, and overexpression of miR-143 in brown adipocytes by lentivirus could rescue the enhancement of UCP1 protein expression caused by LPS, hinting miR-143 may be an important regulator of the thermogenesis in brown adipocytes. More importantly, the knockout of miR-143 further enhanced the LPS-induced increase of body temperature and BAT thermogenesis, and this result was further confirmed by in vitro experiments by using primary brown adipocytes. Mechanistically, adenylate cyclase 9 (AC9) is a new target gene of miR-143 and LPS increases BAT thermogenesis by a way of inhibiting miR-143 expression, a negative regulator for AC9. Our study considerably improves our collective understanding of the important function of miR-143 in inflammatory BAT thermogenesis.

## 1. Introduction

The presence of fever is a hallmark of infection and an important part of the immune system. Febrile temperatures are so closely related to inflammatory response that heat is one of the four main signs of inflammation, along with pain, redness and swelling [1]. Fever is triggered by microbial products, among which the most studied is lipopolysaccharide (LPS) of Gram-negative bacteria. Mechanistically, the regulatory pathway of fever is controlled by the sympathetic nervous system (SNS), which secretes norepinephrine and other catecholamines to elevate body temperature by increasing thermogenesis in the brown adipose tissue (BAT) and also by inducing vasoconstriction to prevent passive heat loss [2,3,4]. In the evolution of vertebrates, fever has been persistent, which strongly proves that high temperature is of survival advantage. There is mounting evidence that an increase of 1 °C to 4 °C in body temperature during fever is associated with improved survival and various infection resolutions [5,6,7,8]. However, fever is not universally beneficial, especially in cases of extreme inflammation, where lowering body temperature rather than raising it has evolved as a protective mechanism [9,10]. As previously reported, a low dose of LPS stimulants can cause fever, while a high dose of LPS stimulants leads to hypothermia [9,11].

As a secretory organ, BAT secretes a large number of batokines through autocrine or paracrine actions that lead to enhanced or inhibited thermogenic activity [12,13]. Moreover, there is increasing evidence that inflammatory cytokines (such as IL-1β, IL-27 and IL-33) directly alter the thermogenic activity of adipocytes by regulating fatty acid metabolism, mitochondrial function or group 2 innate lymphocyte function [14,15,16,17]. Inflammatory cytokines can be divided into pyrogenic cytokines and anti-pyrogenic cytokines, the balance between which decides the thermogenic outcome of systemic inflammation [11]. Although BAT thermogenesis is potently influenced by systemic inflammation [11,18,19,20], it remains unclear whether there are other regulatory mechanisms besides hormones secreted by the SNS, in particular, the role of *IL-6* (secreted by brown adipocytes).

MicroRNAs (miRNAs) are endogenous, noncoding small RNAs that post-transcriptionally regulate gene expression and have been shown to play important roles in glucose and lipid metabolism [21,22]. Recent studies have demonstrated that the miRNAs in the BAT is crucial in regulating the pathways that control a range of processes, including adipocyte differentiation, adipogenesis, and thermogenesis [23,24]. As reported, miR-143 is highly expressed in the white adipose tissue (WAT) and BAT [25,26], and miR-143 knockout (KO) mice exhibited stronger ability to maintain body temperature under cold exposure [27], and can effectively resist diet-induced obesity by promoting BAT thermogenesis and inhibiting WAT adipogenesis [28]. More importantly, miR-143 may be involved in the response of adipocytes to obesity-induced macrophage infiltration and cytokines secretion, as treatment of *TNFα* for 24 h to differentiated 3T3-L1 adipocytes reduced the expression of miR-143 [29]. It is also reported that the fever-like temperatures could up-regulate the level of miR-143 in THP-1-derived macrophages and peripheral blood mononuclear cells [30]. However, the role of miR-143 in the fever-induced thermogenesis of brown adipocytes is largely unknown.

Interestingly, there is still debate on whether BAT is involved in infection-induced fever (LPS stimulation). There are reports that LPS promotes brown adipocyte thermogenesis [31], but there are also reports that infection-induced fever is independent of brown adipocytes [32] or suppresses BAT thermogenesis [33]. Therefore, we firstly examined the functional changes of BAT during LPS-induced fever. We found that there was a significant negative correlation between inflammatory factors (*IL-6* and *TNFα*) and the expression of miR-143 in brown adipocytes. Conclusively, our study found that LPS can directly regulate adipocyte thermogenesis through miR-143/adenylate cyclase 9 (AC9)/UCP1 signaling pathway.

## 2. Results

### 2.1. Low-Dose LPS Induced Fever in Mice

It was previously illustrated that lower doses of LPS typically cause fever [9], and BAT plays a role in infection-induced fever [31]. To verify whether LPS has an effect on the thermogenesis of BAT, we treated mice and brown adipocytes with LPS, respectively. We found that LPS treatment could significantly increase the rectal temperature (Figure 1A) and dorsal interscapular surface temperature (Figure 1B,C) (contribute to fever) of mice, and the mRNA levels of *TNFα*, *IL-6*, *Ucp1*, and *Pgc1α* in the BAT, but did not affect the expression of *Cpt1α* and *Cidea* (Figure 1D,E). LPS also significantly increased the expression of UCP1 protein, which is the marker protein of BAT thermogenesis (Figure 1F,G). Moreover, 0.2 μg/mL LPS can directly stimulate the expression of *Ucp1* mRNA and protein in mouse primary brown adipocytes and increase the levels of *TNFα* and *IL-6* mRNA as well (Figure 1H–J). These results suggested that LPS can directly induce the thermogenesis of brown adipocytes.

### 2.2. miR-143 Was Negatively Correlated with the Expression of Inflammatory Factors in the BAT

Up to now, the relationship between miR-143 and immunomodulatory factors is still unknown, while diet and exogenous pyrogens stimuli potently influence inflammatory thermogenesis [11,20]. So, we measured the expression levels of miR-143, *IL-6* and TNF-α in BAT of mice fed with ND or HFD. Since these data are independent of each other, we can present the experimental data into a graph (including ND/HFD feeding [34]). In the present study, we found that miR-143 expression in the BAT of mice (a population including normal diet (ND)-fed and high fat diet (HFD)-fed mice) exhibited a statistically significant inverse relationship with *TNFα* and *IL-6* expression in the BAT (Figure 2A,B). Furthermore, we also found consistent results in primary brown adipocytes treated with LPS (Figure 2C,D). The results hint that the change of miR-143 may be a correlated event to the expression of an inflammatory factor (especially *IL-6*) in adipocytes induced by LPS.

### 2.3. LPS Increased UCP1 Expression of Brown Adipocytes by Downregulating MiR-143

To verify the role of miR-143 in inflammation-induced BAT thermogenesis, we treated mice and primary brown adipocytes with low doses of LPS. To rule out the possibility that stromal vascular fraction (SVF) cells contribute to the LPS-induced miR-143 expression in BAT, we isolated SVF cells and mature adipocytes from BAT of mice treated with or without LPS. The results showed that low-dose LPS treatment significantly reduced the expression of miR-143 and increased the *Ucp1* mRNA expression in mature adipocytes, and did not affect the expression of miR-143 and *Ucp1* in SVF cells isolated from BAT of mice (Figure 3A). Then, we further verified this finding by using primary brown adipocytes. It was found that 0.1–0.8 μg/mL LPS treatment could significantly reduce the expression of miR-143 and 0.2–0.8 μg/mL LPS treatment could significantly increase the expression of *Ucp1* mRNA (Figure 3B,C). Different LPS treatment duration time could also affect the decline of the miR-143 expression and the increase of the *Ucp1* mRNA expression (Figure 3D,E). Moreover, LPS treatment increased UCP1 protein expression in brown adipocytes (Figure 3F–H), and overexpression (OE) of miR-143 using a lentivirus in brown adipocytes significantly inhibited the expression of the UCP1 protein and rescued the effect of LPS on UCP1 protein expression (Figure 3G–I). Therefore, our results showed that LPS promoted BAT thermogenesis through downregulating miR-143.

### 2.4. IL-6 Increased Ucp1 mRNA Expression of Brown Adipocytes by Downregulating miR-143

To further determine the relationship of miR-143 and *IL-6*, we treated brown adipocytes with *IL-6* protein and LMT-28 (an *IL-6* signaling pathway inhibitor). Results showed that 100 ng/mL and 500 ng/mL *IL-6* could significantly reduce the expression of miR-143 and 100 ng/mL *IL-6* could significantly increase the expression of *Ucp1* mRNA (Figure 4A,B). Different *IL-6* treatment duration time could also affect the decline of the miR-143 expression. Noteworthy, we observed significant up-regulation of *Ucp1* mRNA expression at 12 h, which significant down-regulation after 24 h (Figure 3D,E), and it may be due to the excessive inflammation. Moreover, 50 nM and 100 nM LMT could significantly reduce the protein level of phosphorylated STAT3 without affecting the protein level of STAT3 (Figure 4E,F). Interestingly, both LMT and LPS could significantly reduce the expression of miR-143, and the results of LPS + LMT were similar to those of LPS or LMT alone (Figure 2G). More importantly, LMT could significantly inhibit the increase of *Ucp1* and *Pgc1α* mRNA levels induced by LPS but did not affect the levels of *IL-6* and *TNFa* mRNA (Figure 2H). The above results indicated that LPS may inhibited the expression of miR-143 through *IL-6* signaling pathway, thereby promoting the level of *Ucp1* mRNA.

### 2.5. miR-143KO Enhances LPS-Induced Thermogenesis in the BAT

To further explore whether miR-143 is involved in LPS-induced thermogenesis in brown adipocytes, a miR-143 KO mouse model is engaged. Interestingly, LPS led to more robust thermogenesis and dorsal interscapular surface temperature in the BAT of miR-143KO mice, as demonstrated by increased rectal temperature (Figure 5A–C), increased concentrations of *TNFα*, *IL-6* in the BAT (Figure 5D), and also increased thermogenic mRNA expression levels (Figure 5F,G) and UCP1 protein levels in the BAT (Figure 5H,I). And neither thermogenic-related genes [28] and inflammation-related genes (Figure 5E) were significantly different in the BAT of WT and miR-143KO mice without LPS treatment. Moreover, our newly published data showed that miR-143KO did not affect brown adipocyte differentiation, but increased the expression of *TNFα*, *IL-6* and *Ucp1* mRNA (Figure 5J). Moreover, the mRNA expression levels of *Ucp1*, *TNFα* and *IL-6* were also increased in KO primary brown adipocytes without LPS stimuli (Figure 5J). Therefore, miR-143 may affect the thermogenic function of brown adipocytes. At the same time, when stimulated by LPS, miR-143KO further increased the expression of *Ucp1* mRNA and protein in brown adipocytes and promoted the expression of inflammatory factors *TNFα* and *IL-6* mRNA as well (Figure 5K–M). These results also revealed that miR-143 was involved in LPS-induced thermogenesis in the BAT.

### 2.6. miR-143 Targets AC9

To identify the target of miR-143 in thermogenesis, we first employed miRanda and TargetScan target prediction programs, and adenylate cyclase (AC9) was predicted to be a potential target of miR-143 (Figure 6A). We detected the expression profile in tissues and found that the expression of *AC9* mRNA was much higher in BAT than in other tissues, except for the muscle (Figure 6B). We also found that the *AC9* mRNA expression levels in BAT were much higher than in other isoforms (Figure 6C). As expected, AC9 protein expression was significantly increased in the BAT of KO mice compared with control mice fed ND, but there was no significant difference in UCP1 protein (Figure 6D,E). To confirm the direct interaction between miR-143 and the AC9 3′-UTR, a fragment of the AC9-3′-UTR including miR-143 putative recognition sites was inserted downstream of the luciferase gene in the pGL3-Control reporter plasmid. The seed sequence was also mutated or deleted to disrupt miR-143 binding as controls (Figure 6F). An intact miR-143 target site, but not a mutant or deleted miR-143 target site, was able to mediate the repression of reporter gene activity (Figure 6G). More importantly, in vitro trials showed that miR-143OE significantly decreased the AC9 and UCP1 protein expression levels in WT brown adipocytes, and supplementing of miR-143 diminished the elevated expression of AC9 and UCP1 protein in KO brown adipocytes (Figure 6H–J). The evidence above strongly suggested that miR-143 targets AC9.

### 2.7. LPS Regulates Ucp1 Expression through the miR-143-AC9 Signaling Pathway

We further detected the changes of AC9 along with LPS-induced thermogenesis in brown adipocytes. LPS and miR-143KO could significantly increase the expression of *AC9* mRNA and protein and increase the levels of UCP1 protein in brown adipocytes (Figure 7A–C). Moreover, the expressions of *AC9* mRNA and protein in miR-143KO brown adipocytes treated with LPS was further increased (Figure 7A–C). We also found consistent results in vivo (Figure 7D,E). Thus, the above studies showed that LPS regulates the thermogenesis of brown adipocytes through the miR-143-AC9 signaling pathway.

## 3. Discussion

Thermogenesis is influenced by a variety of environmental factors, and systemic inflammation is one of the major factors [20,35]. Although infection-induced fever can confer a survival advantage in vertebrates, the role of BAT during fever remains largely unknown. LPS is a classic inducer of fever, and referring to the results of previous studies, we established a fever mouse model by intraperitoneal injection of 10 μg/kg LPS. Our study found that intraperitoneal injection of 10 μg/kg LPS could induce the increase of body temperature and promote the mRNA expressions of inflammatory factors in mice (Figure 1A,B). Interestingly, 10 μg/kg of LPS could also increase *Ucp1* mRNA and protein levels in the BAT (Figure 1C–E). The same results were obtained by directly stimulating primary brown adipocytes with 0.2 μg/mL LPS (Figure 1F–H). In addition, our study further found that low and medium dose of LPS (0.2–0.8 μg/mL) could stimulate the expression of *Ucp1* mRNA in brown adipocytes (6–48 h) (Figure 3). A study is consistent with our results, in which 0.1 μg/mL LPS treatment could significantly increase the expression levels of *Ucp1* mRNA and oxygen consumption of brown adipocytes [31]. However, another article reported that 0.1 μg/mL LPS treatment significantly reduced the mRNA levels of thermogenetic related genes and oxygen consumption of brown adipocytes, but did not affect the level of UCP1 protein [33]. Furthermore, Eskilsson’s study showed that BAT thermogenesis was not activated during LPS-induced fever [32]. However, in addition to UCP1-mediated thermogenesis, there are many UCP1-independent thermogenic pathways in BAT. Whether LPS-induced fever has an effect on the UCP1-independent thermogenic pathway in UCP1-KO mice is unclear. Therefore, UCP1-KO in BAT does not affect LPS-induced body temperature elevation, which does not fully represent the independence of BAT thermogenesis from infection-induced fever. Our study further validated the promoting effect of LPS on brown adipocyte thermogenesis. It was also found that BAT directly corresponds to pyrogenic LPS during fever.

Many proinflammatory cytokines such as *TNFα* and *IL-6* are expressed under the induction of obesity or LPS. Our study found that LPS induced fever as well as the thermogenesis of brown adipocytes and BAT, plus increasing the levels of *TNFα* and *IL-6* mRNA. *IL-6* can be considered an important positive regulator of BAT activation [36,37,38], while *TNFα* plays the opposite role [39]. Recently, it has also been found that some other anti-inflammatory cytokines directly affect brown adipocyte thermogenesis, such as IL-27 [15] and IL-10 [40]. Our recently published data shows that miR-143 is involved in the BAT thermogenesis [28]. *TNFα* was reported to decrease the expression of miR-143 in differentiated 3T3-L1 adipocytes [29]. Therefore, we further examined the correlation between miR-143 and inflammatory factors. Our data showed that miR-143KO did not affect the inflammation-related mRNA expression in BAT (Figure 5E). However, miR-143 was negatively correlated with *TNFα* and *IL-6* mRNA expression in the BAT and primary brown adipocytes induced by HFD or LPS (Figure 2).

We further verified whether miR-143 was involved in the LPS-induced thermogenesis of brown adipocytes. It was found that LPS treatment could significantly reduce the expression level of miR-143 in BAT and primary brown adipocytes (Figure 3). More interestingly, the overexpression of miR-143 in brown adipocytes rescued the effect of LPS on UCP1 protein expression (Figure 3), indicating the importance of miR-143 in LPS-induced BAT thermogenesis. *IL-6* was known activator of the thermogenesis and LPS induced the expression of *IL-6* [12,13]. Interestingly, 100 ng/mL *IL-6* reduced the expression of miR-143 and increased the expression of *Ucp1* mRNA (Figure 4). Then, we gendered a miR-143KO mice, and our newly published data shown that the miR-143 was knockout in KO mice, and miR-143KO showed neither phenotypic abnormalities nor energy imbalance in BAT thermogenesis in mice under normal conditions. Moreover, removal of miR-143 further enhanced the LPS-induced elevation of body temperature of miR-143KO mice and the increase of BAT thermogenesis and *IL-6* levels as well (Figure 5). Thus, there is mutual inhibition between miR-143 and *IL-6*. LMT inhibitor treatment did not affect the expression of *Ucp1*, which possibly because *IL-6* mediated thermogenesis did not play a major role under normal conditions. Moreover, LMT inhibitor treatment decreased the expression of miR-143, which possibly because the increased expression of *IL-6* (Figure 4). The role of *IL-6* signaling pathway in the process of thermogenesis without LPS stimulation needs further study. More importantly, inhibition of the *IL-6* signaling pathway reverse the increase of LPS-induced thermogenic gene expression and also reduce the expression of miR-143 (Figure 4), which may be caused by LPS, not LMT. The above results indicate that there is a mutual inhibition between miR-143 and *IL-6*, and LPS promoted BAT thermogenesis by promoting *IL-6* signaling pathway and downregulating miR-143.

Thermogenesis occurs mainly through the activation of the β-adrenergic receptor (AR) on brown adipocytes, which leads to the activation of AC and downstream signaling pathways [41]. AC9 was firstly identified in 1996, and it is widely expressed among tissues [42]. It was divided into a single group because of its unique response mechanism [43]. In the present study, *AC9* mRNA in BAT was firstly found to be much higher than that in other tissues except for the muscle, as well as being much higher than other isoforms. Furthermore, AC9 was confirmed to be a novel target of miR-143, as demonstrated by bioinformatic analysis, dual-luciferase reporter assays, and the AC9 protein analysis of KO mice and overexpression of miR-143 both in WT and KO brown adipocytes (Figure 6). AC9 is a physiologically and clinically relevant effector of β_2_-AR signaling in particular [44,45], and it also regulates cAMP production [45,46]. And it is generally accepted that production of cAMP by adenylate cyclase leads to activation of PKA, p38, and phosphorylation of ATF2 to increase the expression of *Ucp1* mRNA in BAT [47]. Then, we found that LPS increased BAT thermogenesis by a way of inhibiting miR-143 expression, a negative regulator for AC9 (Figure 7).

## 4. Materials and Methods

### 4.1. Experimental Animals

Experiments were performed using homozygous miR-143KO mice and homozygous WT mice bred by miR-143 knockout heterozygotes, as previously reported [27,28,48]. Mice were maintained on standard mouse chow and housed with a 12-h light-dark cycle in controlled temperature (22 °C–24 °C) and humidity (50–65%) conditions with free access to food and water. Eight-week-old male mice were used for in vivo experiment. For diet-induced obesity (DIO) studies, mice were fed a 60% kcal HFD (D12492; Research Diet). Tissues and serum were collected after 6 h of fasting at the end of each experiment, quickly snap-frozen in liquid nitrogen, and then stored at −80°C for further experiments. All of the experimental protocols and methods were approved by the College of Animal Science, South China Agricultural University (SCAU-AEC-2010-0416). All of the experiments were conducted following the “The Instructive Notions with Respect to Caring for Laboratory Animals” issued by the Ministry of Science and Technology of the People’s Republic of China.

### 4.2. LPS Treatment of Mice

Eight-week-old male mice were injected intraperitoneally with saline or LPS (O111:B4, Sigma Aldrich, St. Louis, MO, USA) dissolved in saline at 10 μg/kg/body weight. Rectal temperature was measured with a BAT-Microprobe Thermometer (Physitemp 21 Instruments). Dorsal interscapular surface temperatures were calculated by Thermal Imager (FLIR E60bx).

### 4.3. Isolation and Culture of Primary Brown Adipocytes

Brown adipose tissues from 3- to 5-day-old pups were obtained, and then were minced and digested in PBS containing collagenase type I (1 mg/mL, 17018029; Gibco, Shanghai, China) and CaCl_2_ (3 mM) at 37 °C for 30 min. The tissue suspension was filtered using 100 μm (the upper layer is mature adipocytes, and the lower layer is SVF cells) and 40 μm cell strainers (BD Biosciences, Shanghai, China), and cell pellets were resuspended in DMEM/F12 GlutaMAX (Invitrogen, Shanghai, China) supplemented with 10% fetal bovine serum (FBS) and 1% penicillin/streptomycin and then plated. After 2 days of fusion, SVF cells were induced for cell differentiation with medium containing 10% FBS, 1 μM rosiglitazone (R2408, Sigma Aldrich, St. Louis, MO, USA), 1 μM dexamethasone (D4902, Sigma Aldrich, St. Louis, MO, USA), 5 μg/mL insulin (I0546; Sigma Aldrich, St. Louis, MO, USA), 0.5 mM isobutylmethylxanthine (I7018, Sigma Aldrich, St. Louis, MO, USA), 125 nM indomethacin (I7378, Sigma Aldrich, St. Louis, MO, USA), and 1 nM T3 (T2877, Sigma Aldrich, St. Louis, MO, USA) for 2 days. Next, the cells were maintained in a culture medium containing 10% FBS, 5 μg/mL insulin, and 1 nM T3. The media was changed every 2 days. Cells were harvested on day 6 or 8 of differentiation. Mature adipocytes were treated with LPS or *IL-6* protein (HY-P7063, MedChemExpress, Shanghai, China) for the indicated time. For the inhibition of the LPS or *IL-6* signaling pathway, adipocytes were pretreated with LMT-28 (HY-102084, MedChemExpress) for 1 h. Cells were then collected for further analysis.

### 4.4. Analysis of TNFα and IL-6 Concentration in the BAT

The *TNFα* and *IL-6* concentrations were assayed using a *TNFα* and *IL-6* assay kit (Nanjing Jiancheng Biotechnology Institute) according to the manufacturer’s protocol.

### 4.5. qPCR Analyses

Total mRNA was extracted using TRIzol Reagent (15596-026, Thermo Fisher Scientific, Shanghai, China). After DNase I digestion (2270A, Takara Bio, Kusatsu, Shiga, Japan), a total of 1 μg of total RNA was reverse-transcribed into cDNA using MLV Reverse Transcriptase (M1705, Promega, Madison, WI, USA) and oligo (dT) 18 primer or a specific stem-loop primer for miR-143 (3806, Takara Bio). Real-time PCR was carried out in a STRATAGENE Mx3005P sequence detection system with SYBR Green Master Mix (Promega). Results were normalized to the expression of the housekeeping genes GAPDH, 18S or U6 using the 2^−ΔΔCt^ method. The primer sequences used are presented in Table 1.

### 4.6. Western Blot Analysis

Tissues and cells were lysed in RIPA buffer containing 1 mmol/L PMSF protease inhibitor (P7626, Sigma). The protein concentrations were then measured using a BCA Protein Assay kit (Thermo Fisher Scientific, 23227). Primary antibodies against phospho-STAT3 (Tyr705, #9131S, CST, 1:1000), STAT3 (#12640S, CST, 1:1000), UCP1 (#ab10983, Abcam, 1:2000), β-Tubulin (#2146S, CST, 1:1000) and AC9 (ab191423, Abcam, 1:1000) were used according to the manufacturer’s instructions. Primary antibodies were incubated in blocking buffer at 4 °C overnight. Secondary Alexa antibodies from Life Technologies were then added for 1 h. The protein density was quantified and analyzed using Image J software.

### 4.7. Plasmid Construction, Transfection

Insert sequences of the lentiviral vectors used are presented in Table 2. Lentiviral vectors were produced by transfection of HEK 293T cells in a 6-well plate with 1.5 μg of pMD2.G (Addgene, #12259), 0.5 μg of psPAX2 (Addgene, #12260), 2 μg of pLVX-blank and pLVX-OE (for miR-143 overexpression), using 3 μL of Lipofectamine 2000 (Invitrogen). The media were changed the next day, and the supernatant was collected over the next 48 h and 72 h. The collected media were pelleted at 300× *g* for 4 min, and the supernatant was passed through a 0.45-μm pore size durapore polyvinylidene difluoride (PVDF) membrane (Steriflips; Millipore, Bedford, MA, USA) to remove cellular debris, and then Millipore Amicon-Ultracentrifuge unites (100 kD, 15 mL) were used to concentrate virus. Then, virus stocks were aliquoted into tubes and stored at −80 °C for future use in respective experiments. For primary adipocytes, each type of virus was diluted to the desired virus strength in a fresh medium. The diluted virus was then added to cells overnight. Afterward, the samples were replenished in fresh medium every 2 days. After 3 days, the cells were collected for the following experiments.

### 4.8. Luciferase Reporter Assays

For promoter-luciferase reporter assays, mouse AC9 3′-UTR sequences including miR-143-3p target sites were synthesized (Sangon, Shanghai, China). These synthetic sequences were inserted into the pGL3-Control vector (Promega Co., Madison, WI, USA), downstream of the luciferase gene. Meanwhile, mutagenic and deleted AC9 3′-UTR reporter vectors were constructed with 8 exchanged nucleotides or a deleted target site in a similar way. Differentiated 3T3-L1 cells were seeded in 24 well plates at an appropriate density 1 day before transfection. Then 300 ng of each reporter construct was co-transfected with a miR-143 mimic or negative control (NC). Luciferase assays were performed using a Dual-Luciferase reporter assay system (Promega Co.), with firefly luciferase activity normalized to renilla activity.

### 4.9. Statistical Analyses

All of the results were expressed as the mean ± SEM and analyses were performed using Microsoft Excel or GraphPad Prism 9. A Student’s *t*-test was used for a single variable comparison between two groups. Two-way ANOVA followed by Tukey’s post hoc test was used to examine interactions between multiple variables, and a value of *p* < 0.05 was considered to be statistically significant.

## 5. Conclusions

In conclusion, our study found the expression of IL6 and *TNFα* in brown adipocytes is significantly negatively correlated with miR-143, and the *IL-6* pathway may mediate inhibition of miR-143 expression. Low and medium doses of LPS significantly decreased the expression of miR-143 in brown adipocytes and increased the expression of AC9 and UCP1. Also, miR-143KO further enhanced the BAT thermogenesis induced by LPS. Mechanistically, AC9 is a new target gene of miR-143 and LPS increases BAT thermogenesis by a way of inhibiting miR-143 expression, a negative regulator for AC9. Our study considerably improves our collective understanding of the importance of miR-143 in inflammatory BAT thermogenesis.

## Figures and Tables

**Figure 1 ijms-23-13805-f001:**
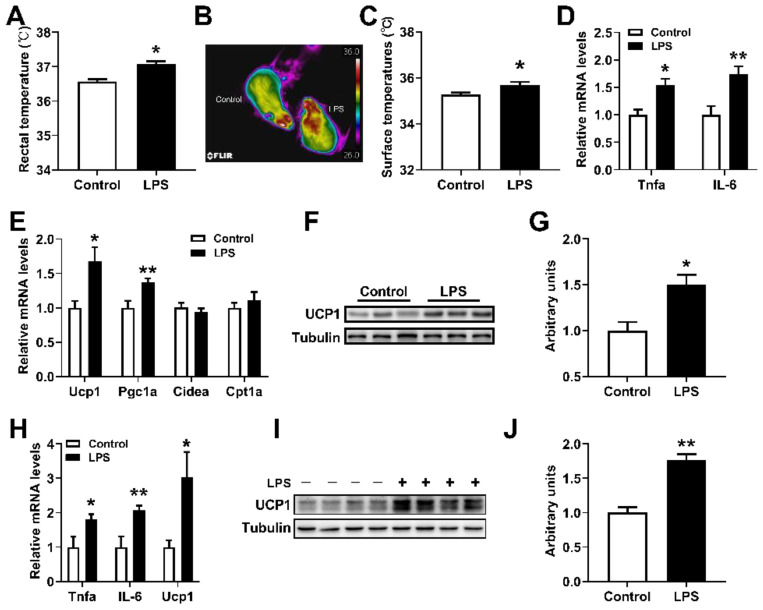
Low-dose LPS induced fever in mice. (**A**) Rectal temperatures of mice treated with 10 μg/kg LPS for 1.5 h. (**B**,**C**) Dorsal interscapular surface imaging and temperature. (**D**,**E**) The inflammation-related (**D**) and thermogenic-related (**E**) mRNA expression levels in the BAT of mice treated with 10 μg/kg LPS for 4 h (n = 6). (**F**,**G**) The UCP1 protein levels in the BAT. (**H**–**J**) The *TNFα*, *IL-6*, *Ucp1* mRNAs (**H**), and UCP1 protein (**I**,**J**) expression levels in primary brown adipocytes treated with or without 0.2 μg/mL LPS. Data are presented as the mean ± SEM. * *p* < 0.05 vs. controls; ** *p* < 0.01 vs. controls, as determined by Two-tailed unpaired Student’s *t*-test.

**Figure 2 ijms-23-13805-f002:**
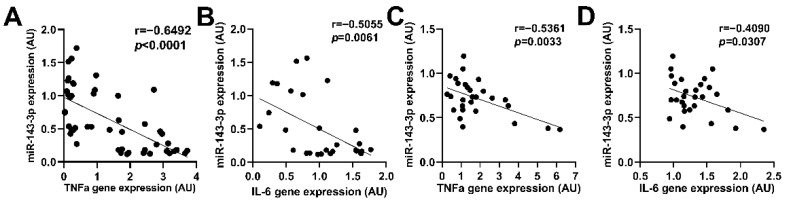
miR-143 was negatively correlated with the expression of inflammatory factors in the BAT. (**A**,**B**) Correlation between miR-143 expression and *TNFα* (**A**) or *IL-6* (**B**) in the BAT of mice fed HFD for 0, 1, 3, 7, 14, 21 days (n = 48 and 28). (**C**,**D**) Correlation between miR-143 expression and *TNFα* (**C**) or *IL-6* (**D**) in primary brown adipocytes treated with 0.2 μg/mL LPS for 0, 12, 24, 48 h (n = 28). The correlation between miR-143 and *IL-6* or *TNFα* was analyzed by GraphPad Prism 9.0.

**Figure 3 ijms-23-13805-f003:**
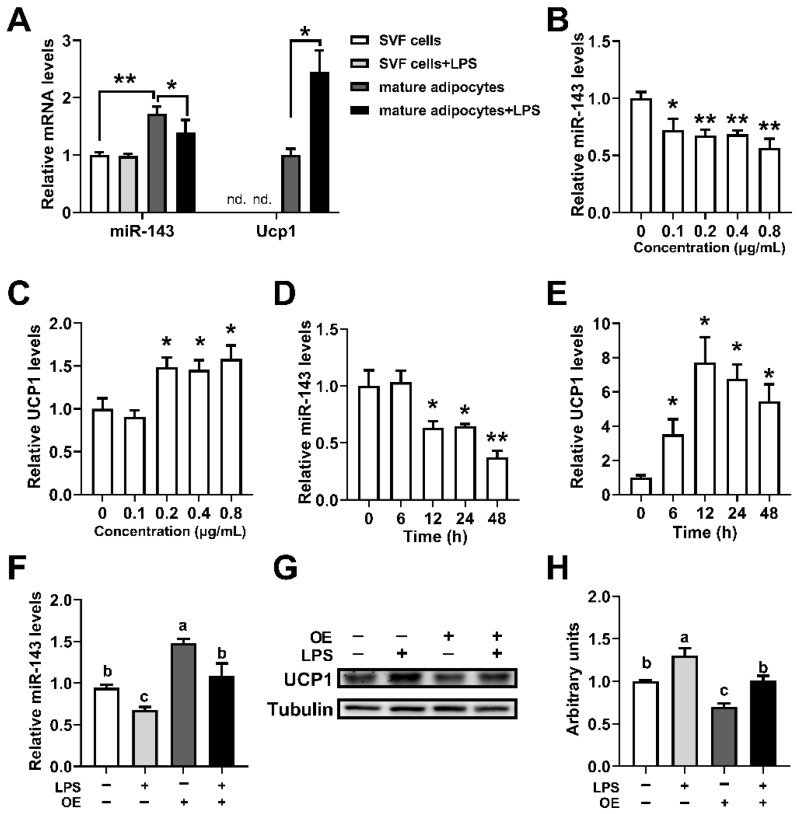
LPS increased UCP1 expression in brown adipocytes while downregulating miR-143. (**A**) The miR-143 and *Ucp1* mRNA expression levels in the isolated SVF cells and mature adipocytes from BAT of mice treated with or without 10 μg/kg LPS (n = 6); nd., not detected. (**B**,**C**) The miR-143 and *Ucp1* mRNA expression in primary brown adipocytes in response to 24 h treatment with various concentrations of LPS (n = 6). (**D**,**E**) Time course of miR-143 and *Ucp1* mRNA induction in primary brown adipocytes in response to LPS (0.2 μg/mL) treatment (n = 6). (**F**–**H**) The miR-143 expression (**G**) and UCP1 protein levels (**H**) in WT and miR-143-overexpressed primary brown adipocytes treated with or without 0.2 μg/mL LPS. Data are presented as the mean ± SEM. * *p* < 0.05 vs. controls; ** *p* < 0.01 vs. controls. Different letters above bars represent significant differences, with shared letters representing no significant differences, as determined by a two-tailed unpaired Student’s *t*-test (**A**–**F**) or two-way ANOVA followed by Tukey’s post hoc test (**G**).

**Figure 4 ijms-23-13805-f004:**
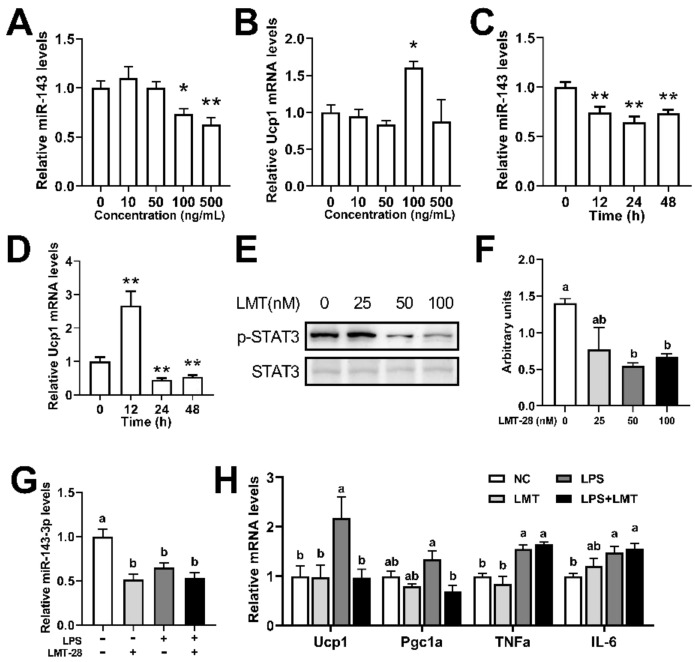
*IL-6* increased *Ucp1* expression in brown adipocytes while downregulating miR-143. (**A**,**B**) The miR-143 and *Ucp1* mRNA expression in primary brown adipocytes in response to 12 h treatment with various concentrations of *IL-6* protein (n = 8). (**C**,**D**) Time course of miR-143 and *Ucp1* mRNA induction in primary brown adipocytes in response to *IL-6* (100 ng/mL) treatment (n = 8). (**E**,**F**) The p-STAT3 and STAT3 protein expression levels in primary brown adipocytes treated with or without LMT for 1 h (n = 3). (**G**,**H**) The miR-143 and *Ucp1*, *Pgc1a*, *TNFa*, and *IL-6* mRNA expression levels in the primary brown adipocytes treated with or without 50 nM LMT for 1h. Data are presented as the mean ± SEM. * *p* < 0.05 vs. controls; ** *p* < 0.01 vs. controls. Different letters above bars represent significant differences, with shared letters representing no significant differences, as determined by a two-tailed unpaired Student’s *t*-test (**A**–**D**) or two-way ANOVA followed by Tukey’s post hoc test (**F**–**H**).

**Figure 5 ijms-23-13805-f005:**
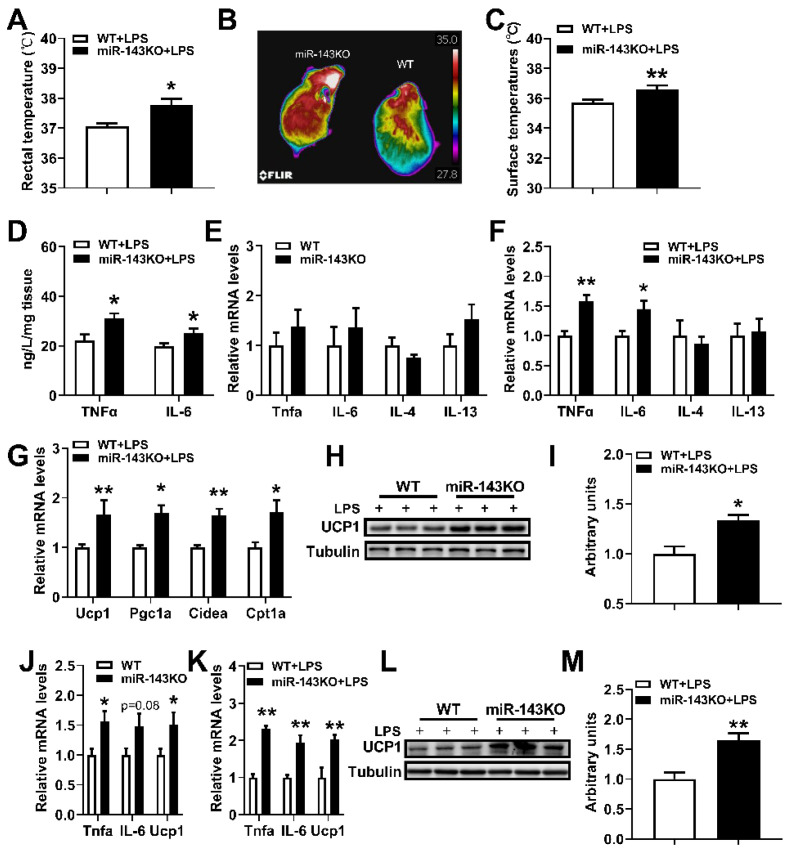
miR-143KO promotes LPS induced thermogenesis in the BAT. (**A**) Rectal temperatures of WT and miR-143KO mice treated with 10 μg/kg LPS for 1.5 h. (**B**,**C**) Dorsal interscapular surface imaging and temperature. (**D**) The concentrations of *TNFα* and *IL-6* in the BAT of mice treated with 10 μg/kg LPS for 4 h (n = 6). (**E**) The inflammation-related mRNA expression levels in the BAT of WT and miR-143KO mice (n = 8). (**F**,**G**) The inflammation-related (**E**) and thermogenic-related (**F**) mRNA expression levels in the BAT of WT and miR-143KO mice treated with 10 μg/kg LPS for 4 h (n = 6). (**H**,**I**) The UCP1 protein levels in the BAT. (**J**) The mRNA expression levels of TNF-α, *IL-6* and UCP1 in the WT and miR-143KO primary brown adipocytes (n = 6). (**K**–**M**) The *TNFα*, *IL-6*, *Ucp1* mRNAs (**J**), and *Ucp1* protein (**K**,**L**) expression levels in WT or miR-143KO primary brown adipocytes treated with 0.2 μg/mL LPS. Data are presented as the mean ± SEM. * *p* < 0.05 vs. controls; ** *p* < 0.01 vs. controls, as determined by Two-tailed unpaired Student’s *t*-test.

**Figure 6 ijms-23-13805-f006:**
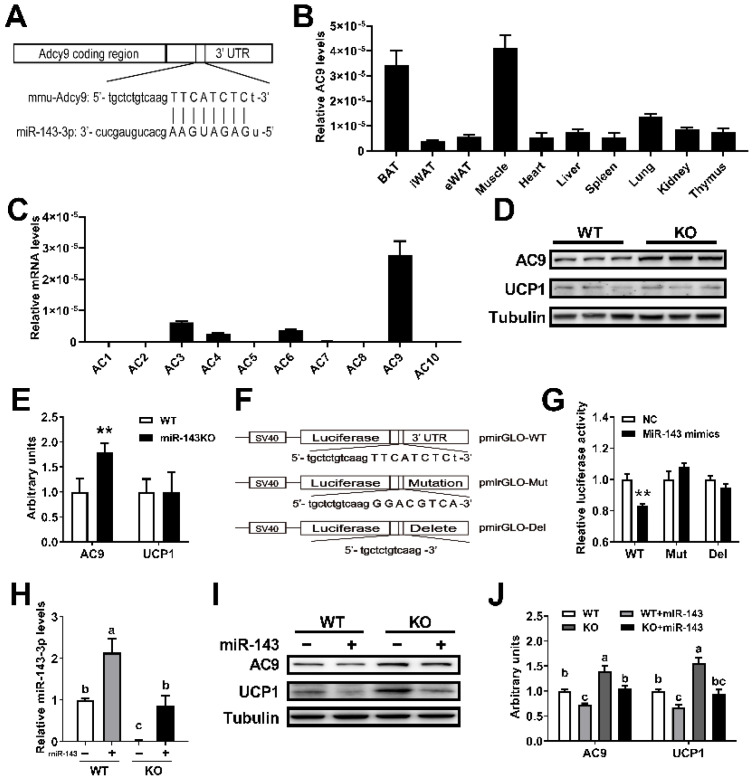
miR-143 targets AC9. (**A**) The predicted target site for miR-143 in the 3′-UTR of AC9. (**B**) The tissue distribution of mRNA encoding *AC9* (n = 6–8). (**C**) The type 1 to 10 adenylyl cyclase mRNA expression levels in BAT (n = 6–8). (**D**,**E**) The AC9 and UCP1 protein expression levels in the BAT of WT and KO mice fed ND (n = 3). (**F**) AC9 3′-UTR with fully changed nucleotides in the seed sequence or with a deleted seed sequence disrupts the binding of miR-143. (**G**) Three 3′-UTR sequences were respectively cloned into the pGL3-Control vector downstream of the luc^+^ gene, and luciferase assays with the constructed vectors were transfected into differentiated 3T3-L1 cells with negative control (NC) or miR-143 mimic were performed (n = 8–10). (**H**–**J**) The miR-143 (**H**) (n = 4) and AC9 protein (**I**,**J**) (n = 3) expression levels in WT or KO brown adipocytes treated with blank or miR-143 expression lentivirus. Data are presented as the mean ± SEM. ** *p* < 0.01 vs. controls. Different letters above groups represent significant differences, with shared letters representing no significant differences, as determined by a two-tailed unpaired Student’s *t*-test (**E**,**G**) and two-way ANOVA followed by Tukey’s post hoc test (**H**,**J**).

**Figure 7 ijms-23-13805-f007:**
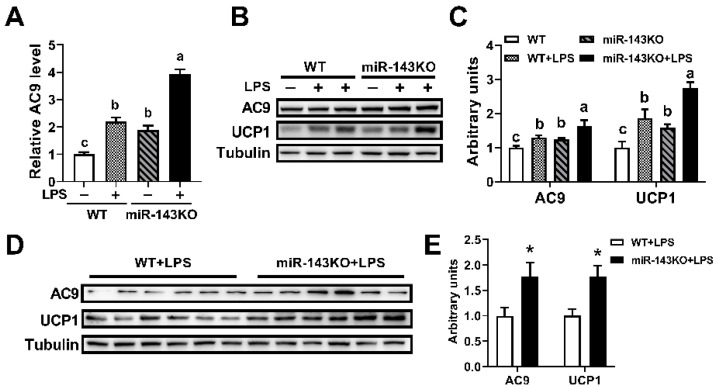
LPS regulates UCP1 expression through the miR-143-AC9 signaling pathway. (**A**–**C**) The *AC9* mRNAs (**A**), and AC9, UCP1 protein (**B**,**C**) expression levels in WT or miR-143KO primary brown adipocytes treated with or without 0.2 μg/mL LPS. (**D**,**E**) The AC9 and UCP1 protein expression levels in mice treated with 10 μg/kg LPS. Data are presented as the mean ± SEM. * *p* < 0.05 vs. controls. Different letters above groups represent significant differences, with shared letters representing no significant differences, as determined by a two-tailed unpaired Student’s *t*-test (**E**) and two-way ANOVA followed by Tukey’s post hoc test (**A**,**C**).

**Table 1 ijms-23-13805-t001:** Primer sequences for quantitative real-time PCR.

Gene	Forward (5′–3′)	Reverse (5′–3′)
mu-Ucp1	ACTGCCACACCTCCAGTCATT	CTTTGCCTCACTCAGGATTGG
mu-Pgc1a	AGCCGTGACCACTGACAACGAG	GCTGCATGGTTCTGAGTGCTAAG
mu-Cidea	ATCACAACTGGCCTGGTTACG	TACTACCCGGTGTCCATTTCT
mu-Cpt1a	CTCCGCCTGAGCCATGAAG	CACCAGTGATGATGCCATTCT
mu-Tnfα	CCTGTAGCCCACGTCGTAGC	AGCAATGACTCCAAAGTAGACC
mu-*IL-6*	AAGTGCATCATCGTTGTTCATAC	CCATCCAGTTGCCTTCTTG
mu-IL-4	ATCATCGGCATTTTGAACGAGG	TGCAGCTCCATGAGAACACTA
mu-IL-13	TGAGCAACATCACACAAGACC	GGCCTTGCGGTTACAGAGG
mu-U6	CTCGCTTCGGCAGCACA	AACGCTTCACGAATTTGCGT
mu-18s	CTTAGTTGGTGGAGCGATTT	GCTGAACGCCACTTGTCC
mu-AC1	GTCACCTTCGTGTCCTATGCC	TTCACACCAAAGAAGAGCAGG
mu-AC2	GACTGGCTCTACGAGTCCTAC	GGGCAGTGGGAACGGTTAT
mu-AC3	CTCGCTTTATGCGGCTGAC	ACATCACTACCACGTAGCAGT
mu-AC4	AGTACCCACTGCTGATACTGC	AGCCACCCAAAGCACACAG
mu-AC5	CTTGGGGAGAAGCCGATTCC	ACCGCTTAGTGGAGGGTCT
mu-AC6	GATGAACGGAAAACAGCTTGGG	GGTGGCTCCGCATTCTTGA
mu-AC7	AAGGGGCGCTACTTCCTAAAT	GTGTCTGCGGAGATCCTCA
mu-AC8	CTCTACACCATCCAACCGACG	GCACCGAGTCGCTAGACAG
mu-AC9	ACCTTCTTCCTCCTGCTCCTCTTG	GATGATGTTCCGCAGTAGCCAGTC
mu-AC10	GGCAGGAATTACAAGACAGGG	GCACTTTCTCCACTATGGCACT
mu-qmiR-143	GGGTGAGATGAAGCACTG	CAGTGCGTGTCGTGGAGT
mu-miR143-RT	GTCGTATCCAGTGCGTGTCGTGGAGTCGGCAATTGCACTGGATACGACGAGCTA	

**Table 2 ijms-23-13805-t002:** Insert sequence in lentiviral vectors.

Gene	Sequences
miR-143 expression-F	AATTCCCTGAGGTGCAGTGCTGCATCTCTGGTCAGTTGGGAGTCTGAGATGAAGCACTGTAGCTCAGGTTTTTTG
miR-143 expression-R	GATCCAAAAAACCTGAGGTGCAGTGCTGCATCTCTGGTCAGTTGGGAGTCTGAGATGAAGCACTGTAGCTCAGG

## Data Availability

Not applicable.

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
