# Peer review of "LPS-Induced Inhibition of miR-143 Expression in Brown Adipocytes Promotes Thermogenesis and Fever"

_ijms, 2022, doi:10.3390/ijms232213805_

Round 1

Reviewer 1 Report (Previous Reviewer 1)

The authors revised the manuscript appropriately.

Author Response

Thanks for your comments.

Reviewer 2 Report (New Reviewer)

Author Response

Part A: Major concerns (comments on scientific content)

  1. In Fig. 2, panels A-D, Could the authors please explain the rationale behind combining data from ND and HFD into a single graph for both TNF-alpha and IL-6 (A and B)? And also for with or without LPS (C and D)?

Response: Thanks for your comments. It is well known that high-fat diet and LPS stimuli increased the expression of inflammatory factors, including TNF-α and IL-6 [1,2]. Up to now, the relationship between miR-143 and immunomodulatory factors is still unknown. So, we measured the expression levels of miR-143, IL-6 and TNF-α in BAT of mice fed with ND or HFD. Since these data are independent of each other, we can present the experimental data into a graph (including ND/HFD feeding [3]), and a large number of data showed that miR-143 was negatively correlated with TNF-α and IL-6. It also applies to the data of LPS treatment.

  1. In Fig. 4, please include a graph showing mRNA levels of miR-143 in different tissues including BAT, to indicate the levels of the whole body knock-out. Also, given the fact that inflammatory cytokines were upregulated, what was the status of other tissues with respect to immune cell infiltration and overall health of the mice? Could the authors please elaborate on this?

Response: Thanks for your comments. The miR-143 was knockout in KO mice, and miR-143KO showed neither phenotypic abnormalities nor energy imbalance in BAT thermogenesis in mice under normal conditions (as shown below). Unfortunately, like other studies focusing on LPS induced fever [4,5], we did not detect immune cell infiltration in other tissues. However, during the experiment of mice injected with low-dose LPS, except for the increase of body temperature, the mice did not suffer from decreased vitality, diarrhea and death. The following figure has been included in the latest article accepted in the International Journal of Molecular Science (title: miR-143-null is against diet-induced obesity by promoting BAT thermogenesis and inhibiting WAT adipogenesis).

  1. In Fig. 4, panels E and F, please include a sham injection control for both WT and miR-143 KO in the graphs showing the levels of cytokines and thermogenic markers.

Response: Thanks for your comments, neither thermogenic-related genes (As shown in the figure above) and inflammation-related genes (Fig. 4E) were significantly different in the BAT of WT and miR-143KO mice without LPS treatment. Since the WT/KO mice without LPS injection and the WT/KO mice with LPS injection were not completed in the same batch of experiments, we cannot put them in the same table. We apologize for this.

  1. In Fig. 4, panels I-K, please include data from control primary brown adipocytes without any LPS injection.

Response: Thanks for your comments. We detected the genes expression of TNFa, IL-6 and UCP1 in WT and KO primary brown adipocytes (Fig. 5J). And the protein expression of UCP1 in WT and KO primary brown adipocytes treated with or without LPS are shown in Figure 7.

  1. In Fig. 6, panels D-E, please include a sham control i.e. without LPS injection. Also include levels of UCP1 for sham, LPS injection and LPS + miR143 KO. Additionally, can the authors show if the AC9 is sufficient and/or necessary for UCP1 induction through LPS? There can be other ways of inducing UCP1 as well. For example, doing a knockdown or a knock-out of AC9, and then treating the cells with LPS or miR-143 knockout/over-expression to see the effect on UCP1 levels. There could be other targets of miR-143 as well, and to prove that AC9 is the most significant of them all, additional experiments (as the one suggested above) seem necessary.

Response: Thanks for your comments. The result of AC9 and UCP1 protein expression levels in the BAT of WT and KO mice without LPS treatment were found in Fig. 5D-E. We also detected the protein expression levels of UCP1 (Fig. 7D–E). Additionally, we have demonstrated that AC9 is necessary for thermogenesis (as shown below). This part of the content has been included in the article submitted to the International Journal of Biological Macromolecules 5 months ago, but it is still under review.

Figure shRNA against AC9 reverses the thermogenesis induced by miR-143 deletion

(A–C) The AC9 and UCP1 protein (A–B) (n = 3) and thermogenic-related gene mRNA (C) (n = 6) expression levels in primary brown adipocytes transfected with blank lentivirus or AC9i lentivirus from WT or KO mice. (D) The cAMP concentration of primary brown adipocytes transfected with blank lentivirus or AC9i lentivirus in WT or KO mice (n = 6). (E) Oxygen consumption of primary brown adipocytes transfected with blank lentivirus or AC9i lentivirus in WT or KO mice (n = 8). (F) Rectal temperature of KO mice injected with blank or AC9i lentivirus followed by cold exposure at 10°C for 6 h (n = 8). (G–H) Dorsal interscapular surface imaging and temperature (n = 4). (I–K) The AC9 and UCP1 protein (H–I) (n = 3) and thermogenic-related gene mRNA (J) (n = 6) expression levels in the BAT of KO mice injected with blank or AC9i lentivirus followed by cold exposure to 10°C for 6 h. Data are presented as the mean ± SEM. *p < 0.05 vs controls; **p < 0.01 vs controls. Different letters above groups represent significance differences, with shared letters representing no significant differences, as determined by a two-tailed unpaired Student’s t-test (F–K) and two-way ANOVA followed by Tukey’s post hoc test (B–E).

  1. In discussion, line no. 261, the authors have referred to IL-10 as an “inflammatory” cytokine. However, it is perhaps widely known that IL-10 is in fact an “anti-inflammatory” cytokine.

Response: Thanks for your comments. We have corrected the description in the discussion.

  1. In discussion, lines 275-276, the authors claim that the IL-6 pathway may mediate the inhibition of the miR-143. However, the data from 2 G show that the IL-6 pathway inhibitor LMT in fact reduces the miR-143 levels. Hence, this statement in discussion seems to be in contradiction with the results shown in Fig. 2 G. If the LMT treatment causes the miR-143 to go down, then the PGC-1α/UCP1 levels should go up, unless there is another mechanism keeping the UCP1 levels down. The effects of LMT on miR-143 and UCP1 and PGC-1 alpha could be independent of each other. Could the authors please include an explanation for this?

Response: Thanks for your comments. To further test our view, we treated primary brown adipocytes with or without IL-6 protein. The results showed that 100 ng/mL IL-6 could significantly reduce the expression of miR-143 and increase the mRNA expression of Ucp1 (Fig. 4). Moreover, miR-143KO enhanced the IL-6 and UCP1 levels (Fig. 5). Thus, there is mutual inhibition between miR-143 and IL-6. LMT inhibitor treatment did not affect the expression of Ucp1, which possibly because IL-6 mediated thermogenesis did not play a major role under normal conditions. Moreover, LMT inhibitor treatment decreased the expression of miR-143, which possibly due to the increased expression of IL-6 (Fig. 4). The role of IL-6 signaling pathway in the process of thermogenesis without LPS stimulation needs further study. More importantly, inhibition of the IL-6 signaling pathway reverse the increase of LPS-induced thermogenic gene expression and also reduce the expression of miR-143 (Fig. 4), which may be caused by LPS, not LMT. The above results indicate that there is a mutual inhibition between miR-143 and IL-6, and LPS promoted BAT thermogenesis by promoting IL-6 signaling pathway and downregulating miR-143.

Part B: Minor concerns (comments on presentation and language style)

  1. In lines 70, 72, 82 and 252, the phrase “infection-induced” might be more appropriate than “immune-induced”.

Response: Thanks for your suggestion. We have corrected the description.

  1. 5, please make all the panels a little bigger to make it easier to read.

Response: Thanks for your suggestion. We have modified the display of Fig 6 in main text.

  1. Solinas, G. Molecular pathways linking metabolic inflammation and thermogenesis. Obes Rev 2012, 13, 69–82, doi:10.1111/j.1467-789X.2012.01047.x.
  2. Stock, M.J. Gluttony and thermogenesis revisited. Int J Obes Relat Metab Disord 1999, 23, 1105-1117, doi:10.1038/sj.ijo.0801108.
  3. Yuan, Y.; Zhu, C.; Wang, Y.; Sun, J.; Feng, J.; Ma, Z.; Li, P.; Peng, W.; Yin, C.; Xu, G.; et al. alpha-Ketoglutaric acid ameliorates hyperglycemia in diabetes by inhibiting hepatic gluconeogenesis via serpina1e signaling. Sci Adv 2022, 8, eabn2879, doi:10.1126/sciadv.abn2879.
  4. Romanovsky AA, S.O., Sakurada S, Sugimoto N, Nagasaka T. Endotoxin shock thermoregulatory mechanisms. Am J Physiol-Reg I 1996, 270, R693–R703.
  5. Munro, P.; Dufies, O.; Rekima, S.; Loubat, A.; Duranton, C.; Boyer, L.; Pisani, D.F. Modulation of the inflammatory response to LPS by the recruitment and activation of brown and brite adipocytes in mice. Am J Physiol-Endoc M 2020, 319, E912-E922, doi:10.1152/ajpendo.00279.2020.

Round 2

Reviewer 2 Report (New Reviewer)

The authors have addressed all of the previously reported concerns. Please refer to the attached document.

Author Response

Reviewer response: The authors’ response is satisfactory. Could the authors please include this explanation somewhere in the results or discussion?

Response: Thank you for your suggestion. We added this explanation to the results (line 106–108).

Reviewer response: The authors’ response is satisfactory. Could the authors please cite this manuscript and briefly mention the KO was reported previously?

Response: Thank you for your suggestion. In the Introduction, we added a description of miR-143KO mice and a reference to our newly published article (line 62–65).

This manuscript is a resubmission of an earlier submission. The following is a list of the peer review reports and author responses from that submission.

Round 1

Reviewer 1 Report

In the present study, Liu et al. found that LPS promotes BAT thermogenesis with downregulation of miR-143. They also suggested that IL6 signaling pathway may be involved in the inhibition of miR-143. Although the authors did not show the direct evidence, they proposed that miR-143 regulates thermogenesis via the adenylate cyclase 9-PKA-UCP1 pathway. The findings are potentially interesting, but there are several questions that deflate the value of the present study.

Major criticisms

 The authors described that “Our unpublished study found that miR-143 directly targeted AC9 and regulated the thermogenesis of brown adipocytes through the AC9-PKA-AMPK signaling pathway.” The authors must show the experimental evidence of the involvement of the signaling pathway in the present study. It is generally accepted that production of cAMP by adenylate cyclase leads to activation of PKA, p38, and phosphorylation of ATF2 to increase the expression of UCP1 mRNA in BAT. Is AC9 selectively expressed in BAT? How does miR-143 regulate the expression of UCP1? Does miR-143 bind to the 3’UTR of AC9 mRNA to inhibit the expression? Please show the importance of PKA-AMPK-UCP1 pathway. The authors must show the mechanisms with experimental data.

Wong et al. reported that miR-143 is a temperature-sensitive miRNA and high temperature increases the expression of miR-143 (NAR 2016). On the other hand, the authors indicated that miR-143 expression is decreased in the high temperature conditions (Figures 1a and 3a). Is the regulation of miR-143 by temperature different in cells (brown adipocytes and THP1 cells)?

 miR-143 is shown to inhibit the expression of IL6 and TNFa (NAR 2016). What is the mechanism of IL6-mediated inhibition of miR-143.

Author Response

  1. The authors described that “Our unpublished study found that miR-143 directly targeted AC9 and regulated the thermogenesis of brown adipocytes through the AC9-PKA-AMPK signaling pathway.” The authors must show the experimental evidence of the involvement of the signaling pathway in the present study. It is generally accepted that production of cAMP by adenylate cyclase leads to activation of PKA, p38, and phosphorylation of ATF2 to increase the expression of UCP1 mRNA in BAT. Is AC9 selectively expressed in BAT? How does miR-143 regulate the expression of UCP1? Does miR-143 bind to the 3’UTR of AC9 mRNA to inhibit the expression? Please show the importance of PKA-AMPK-UCP1 pathway. The authors must show the mechanisms with experimental data. Response: Thanks for your question. This question is the key point to understand function of miR-143, and it was well illustrated in another article submitted to the International Journal of Biological Macromolecules 4 months ago, which is still under review. We have detected the expression profile in tissues and found that the expression of AC9 mRNA was much higher in BAT than in other tissues (Fig. 1A). We also found that the AC9 mRNA expression level in BAT was much higher than that in other isoforms (Fig. 1B). Then we employed miRanda and TargetScan target prediction programs, and AC9 was predicted to be a potential target of miR-143 (Fig. 2A). AC9 protein expression was significantly increased in the BAT of KO mice compared with control mice fed ND (Fig. 2B–C). To confirm the direct interaction between miR-143 and the AC9 3’-UTR, a fragment of the AC9-3’-UTR including miR-143 putative recognition sites was inserted downstream of the luciferase gene in the pGL3-Control reporter plasmid. The seed sequence was also mutated or deleted to disrupt miR-143 binding as controls (Fig. 2D). An intact miR-143 target site, but not a mutant or deleted miR-143 target site, was able to mediate the repression of reporter gene activity (Fig. 2E). More importantly, in vitro trials showed that miR-143 OE significantly decreased the AC9 and UCP1 protein expression levels in WT brown adipocytes (Fig. 2F–H), and supplementing of miR-143 diminished the elevated expression of AC9 and UCP1 protein in KO brown adipocytes (Fig. 2F–H). And supplementing of miR-143 in BAT of cold exposed KO mice also diminished the elevated expression of AC9 and UCP1 protein in BAT of KO mice (Fig. 2E–F). The evidence above strongly suggested that miR-143 directly targets AC9. According to this suggestion, we revised our abstract (line 25–26) and conclusive description (line 264–266). 
  2. Wong et al. reported that miR-143 is a temperature-sensitive miRNA and high temperature increases the expression of miR-143 (NAR 2016). On the other hand, the authors indicated that miR-143 expression is decreased in the high temperature conditions (Figures 1a and 3a). Is the regulation of miR-143 by temperature different in cells (brown adipocytes and THP1 cells)? Response: Thanks for your question. It is well known low-dose LPS increased body temperature. In our study, low-dose LPS was found to decrease miR143 expression. Results above are understandable since miR-143 is a negative factor in BAT thermogenesis. Although macrophages are also involved in the regulation of body temperature, the way they regulate body temperature is completely different from that of brown adipocytes. Wong et al. reported that the expression of miR-143 was elevated in THP-1-derived macrophages at higher ambient temperatures, thereby fine-reducing the expression of endogenous pyrogens, which in turn leads to decreased body temperature. Our study shows the effect of LPS stimulation (but not high temperature) on miR-143 in brown adipocytes. Therefore, the regulation of miR-143 by temperature different in cells (brown adipocytes and THP-1-derived macrophages) is remains unclear.
  3. miR-143 is shown to inhibit the expression of IL6 and TNFa (NAR 2016). What is the mechanism of IL6-mediated inhibition of miR-143. Response: Thanks for your question. We found that inhibiting the IL-6 signaling pathway inhibited Ucp1 mRNA expression in brown adipocytes, and also inhibited the expression of miR-143. We speculate that the downstream proteins of the IL-6 signaling pathway have negative regulation on the expression of miR-143. So far, there are no reports to explore the inhibitory effect of IL-6 on miR-143. Thus, the mechanism of IL-6-mediated inhibition of miR-143 expression requires further study.

Reviewer 2 Report

In the manuscript entitled “Inhibition of miR-143 expression in brown adipocytes by LPS promotes thermogenesis and fever” the author investigated the potential roles of miR-143 in low-dose LPS-induced thermogenesis in brown adipocytes and demonstrated that low-dose LPS treatment could increase UCP1 while downregulating miR-143 expression in brown adipocytes. They further concluded that LPS promotes the adenylate cyclase9-UCP1 signaling pathway to regulate the thermogenesis of brown adipocytes by reducing the miR-143 level. The finding is interesting, but some of the conclusions are overstated. This manuscript also suffers from some other problems that detract from the overall quality. I have some comments shown below to help improve the quality of this manuscript.    

  1. The conclusion in Line 21 to Line 22 “Mechanistically, LPS promotes the adenylate cyclase9-UCP1 signaling pathway to regulate the thermogenesis of brown adipocytes by reducing the level of miR143” is overstated, which hasn’t been demonstrated in the manuscript.    
  2. It would be better to include the previous studies about the function of BAT in LPS-induced fever (Munro, Patrick, AJP, 2020 and Eskilsson, Anna, FASEB, 2020) in the introduction.    
  3. The author cited the reference (Eskilsson, Anna, FASEB, 2020) in Line 79 to Line 80 to highlight the role of BAT in immune-induced fever, however, the major conclusion in that paper is that “BAT thermogenesis is not activated during LPS-induced fever”. Please justify and discuss in the “Discussion”.    
  4. For the WB in Figure 1, Figure 3, Figure 4, and Figure 5, it would be better to use the housekeeping protein, such as b-actin as the loading control.    
  5. For Figure 1A and Figure 4A, does BAT temperature change after low LPS treatment?     
  6. In Figure 1 and Figure 2, the primary brown adipocytes used in this manuscript are in vitro differentiated SVF cells, which contain different cell types. Do SVF cells from BAT can express miR143? It would be better to use the SVF cells as the control.    
  7. Figure 2F is inconsistent with that shown in the “Results”. It would be better to add the quantification for Figure 2E.    
  8. For Figure 3G, Figure 3I, and Figure 4C, it would be better to annotate what “a”, “b”, and “c” are present in the figure legends.    
  9. Figure legends for Figure 4H and 4I in line 168 are inconsistent with that shown in the Figure.    
  10. In Figure 4, Do thermogenic-related genes and inflammation-related genes change in the BAT of WT and miR-143 KO mice without LPS treatment?    
  11. As Figure 5B showed increased UCP1 expression in miR-143 KO brown adipocyte without LPS treatment, does miR-143 knockout or overexpression affect the adipogenesis of BAT svf?    
  12. In Figure 5A and Figure 5B, LPS treatment can induce AC9 and UCP1 levels in miR-143 KO brown adipocyte, which indicated that miR-143 is dispensable for LPS-induced thermogenesis, which is contradictory to the conclusion in this manuscript. Please provide an explanation.   
  13. The quantification of UCP1 expression in Figure 5C (WT vs WT+LPS) is inconsistent with that in Figure 5B. please justify.   
  14. The conclusion in Line 179 to Line 180 ”Thus, the above studies show that LPS regulates the thermogenesis of brown adipocytes through the miR-143-AC signaling pathway” was overstated. To get such a conclusion, the author should investigate whether inhibition of AC9-PKA-AMPK signaling could attenuate LPS-induced thermogenesis.   
  15. In “4.3. Isolation and culture of primary brown adipocytes” of the “Materials and Method”, it would be better to replace the preadipocytes with the stromal vascular fraction (SVF), as the author can only get SVF cells (heterogeneous mixture of different cell types) by using the method described in the manuscript.  

Author Response

  • The conclusion in Line 21 to Line 22 “Mechanistically, LPS promotes the adenylate cyclase9-UCP1 signaling pathway to regulate the thermogenesis of brown adipocytes by reducing the level of miR143” is overstated, which hasn’t been demonstrated in the manuscript.

Response: Thank you for your reminder. As the answer to question1 of the first reviewer, this question was well illustrated in another article submitted to the International Journal of Biological Macromolecules 4 months ago, which is still under review. As shown in Fig. 1, we employed miRanda and TargetScan target prediction programs, and adenylate cyclase (AC9) was predicted to be a potential target of miR-143 (Fig. 1A). AC9 protein expression was significantly increased in the BAT of KO mice compared with control mice fed ND (Fig. 1B–C). To confirm the direct interaction between miR-143 and the AC9 3’-UTR, a fragment of the AC9-3’-UTR including miR-143 putative recognition sites was inserted downstream of the luciferase gene in the pGL3-Control reporter plasmid. The seed sequence was also mutated or deleted to disrupt miR-143 binding as controls (Fig. 1D). An intact miR-143 target site, but not a mutant or deleted miR-143 target site, was able to mediate the repression of reporter gene activity (Fig. 1E). More importantly, in vitro trials showed that miR-143 OE significantly decreased the AC9 and UCP1 protein expression levels in WT brown adipocytes (Fig. 1F–H), and supplementing of miR-143 diminished the elevated expression of AC9 and UCP1 protein in KO brown adipocytes (Fig. 1F–H). And supplementing of miR-143 in BAT of cold exposed KO mice also diminished the elevated expression of AC9 and UCP1 protein in BAT of KO mice (Fig. 1E–F). The evidence above strongly suggested that miR-143 directly targets AC9. 

  • It would be better to include the previous studies about the function of BAT in LPS-induced fever (Munro, Patrick, AJP, 2020 and Eskilsson, Anna, FASEB, 2020) in the introduction.

Response: Thank you for your suggestion. We have raised the issue of inconsistent results from studies on LPS-induced brown adipocyte thermogenesis in the introduction (lines 74–77).

  • The author cited the reference (Eskilsson, Anna, FASEB, 2020) in Line 79 to Line 80 to highlight the role of BAT in immune-induced fever, however, the major conclusion in that paper is that “BAT thermogenesis is not activated during LPS-induced fever”. Please justify and discuss in the “Discussion”.

Response: Thank you for your suggestion. We have changed citations in Line 79. In addition to UCP1-mediated thermogenesis, there are many UCP1-independent thermogenic pathways in BAT. Whether LPS-induced fever has an effect on the UCP1-independent thermogenic pathway in UCP1-KO mice is unclear, therefore, UCP1-KO cannot fully explain the independence of BAT thermogenesis from immune-induced fever. We've discussed this in the Discussion section on lines 220-227.

  • For the WB in Figure 1, Figure 3, Figure 4, and Figure 5, it would be better to use the housekeeping protein, such as b-actin as the loading control.

Response: Thank you for your suggestion. We have replaced ponceau S with tubulin.

  • For Figure 1A and Figure 4A, does BAT temperature change after low LPS treatment?

Response: Thank you for pointing out this problem. We examined BAT temperature in mice and found that they showed the same trend as body temperature, and the results are shown in Figure 1B-C and Figure 4B-C in the main text.

  • In Figure 1 and Figure 2, the primary brown adipocytes used in this manuscript are in vitro differentiated SVF cells, which contain different cell types. Do SVF cells from BAT can express miR143? It would be better to use the SVF cells as the control.

Response: Thanks for your question. We did not detect the expression of miR-143 in SVF. However, another study reported the changes of miR-143 in the process of adipose tissue-derived stromal cell differentiation (1). It can be speculated that SVF cells expresses miR-143. SVF cells have a good differentiation efficiency in our study, and the differentiated primary brown adipocytes is absolutely dominant (Fig. 3). The main purpose of this paper was to investigate the relationship between LPS-induced fever and brown adipocyte thermogenesis. Therefore, differentiated primary brown adipocytes are more suitable controls than SVF cells.

  • Figure 2F is inconsistent with that shown in the “Results”. It would be better to add the quantification for Figure 2E.

Response: Thanks for your reminder, the 25nM LMT treatment group had no significant difference from the untreated group due to the large intra-group difference. Fig. 2F is the quantification of Fig. 2E, the original abscissa was wrongly identified, which has been changed.

  • For Figure 3G, Figure 3I, and Figure 4C, it would be better to annotate what “a”, “b”, and “c” are present in the figure legends.

Response: Thanks for your suggestion. For Figure 3G and 3I, the meanings of “a”, “b”, and “c” have been defined in the figure legends, in lines 157-160 (different letters above groups represent significant differences, with shared letters representing no significant differences, as determined by two-way ANOVA followed by Tukey’s post hoc test). For Figure 4C, the differences are not represented by “a”, “b”, and “c”, but “*”.

  • Figure legends for Figure 4H and 4I in line 168 are inconsistent with that shown in the Figure.

Response: Thanks for your reminder, we have revised the description in line 181.

  • In Figure 4, Do thermogenic-related genes and inflammation-related genes change in the BAT of WT and miR-143 KO mice without LPS treatment?

Response: Thanks for your question, neither thermogenic-related genes and inflammation-related genes were significantly different in the BAT of WT and miR-143KO mice under normal diet (Fig. 2). This part of the content has been included in the article submitted to the International Journal of Molecular Sciences, but it is still under review.

  • As Figure 5B showed increased UCP1 expression in miR-143 KO brown adipocyte without LPS treatment, does miR-143 knockout or overexpression affect the adipogenesis of BAT svf?

Response: Thanks for your question, we found that miR-143 knockout did not affect brown adipocyte differentiation. The results are shown in the figure below. This part of the content is also included in the article submitted to the International Journal of Molecular Sciences, but it is still under review.

  • In Figure 5A and Figure 5B, LPS treatment can induce AC9 and UCP1 levels in miR-143 KO brown adipocyte, which indicated that miR-143 is dispensable for LPS-induced thermogenesis, which is contradictory to the conclusion in this manuscript. Please provide an explanation.

Response: Our study found that in brown adipocytes, miR-143 inhibited thermogenesis by down-regulating UCP1(main text Fig. 3G–I), and LPS inhibited miR-143 expression in order to increase thermogenesis. miR-143KO diminished negative regulation of UCP1, thus LPS stimulation may further increase the thermogenesis of brown adipocytes. It is certain that that miR-143 is dispensable for LPS-induced thermogenesis, but is involved in thermogenesis regulation.

  • The quantification of UCP1 expression in Figure 5C (WT vs WT+LPS) is inconsistent with that in Figure 5B. please justify.

Response: Thank you for your question, there is a certain difference in the expression of UCP1 protein within the WT+LPS group, therefore, we replaced a more representative band in Figure 5B.

  • The conclusion in Line 179 to Line 180 ”Thus, the above studies show that LPS regulates the thermogenesis of brown adipocytes through the miR-143-AC signaling pathway” was overstated. To get such a conclusion, the author should investigate whether inhibition of AC9-PKA-AMPK signaling could attenuate LPS-induced thermogenesis.

Response: Thank you for your reminder. The relationship of miR-143 and AC9 has been proved and this part is included in another paper in publishing (shown in Fig. 2 above). According this suggestion, we revised our conclusive description as: Collectively, this study eventually revealed that LPS-induced fever can reduce the miR-143 level of brown adipocytes to promote thermogenesis (line 263–266).

  • In “4.3. Isolation and culture of primary brown adipocytes” of the “Materials and Method”, it would be better to replace the preadipocytes with the stromal vascular fraction (SVF), as the author can only get SVF cells (heterogeneous mixture of different cell types) by using the method described in the manuscript.

Response: We deeply appreciate your suggestion, we have revised in Materials and Methods (line 293–294).

  1. Chen L, Hou J, Ye L, Chen Y, Cui J, Tian W, et al. MicroRNA-143 regulates adipogenesis by modulating the MAP2K5-ERK5 signaling. Sci Rep. 2014;4:3819.

Round 2

Reviewer 1 Report

Regulation of AC9 by miR-143: The authors did not improve the revised manuscript according to this reviewer's comments. These results should be included in this manuscript. The contribution of AMPK (line 261) is still unclear.

The regulation of miR-143 by IL6 should be discussed in the section of discussion.

Statistical analysis: the meaning of a, b and c is unclear (Figure 2F-H, Figure 3G-I, and Figure 5).

Author Response

  1. Regulation of AC9 by miR-143: The authors did not improve the revised manuscript according to this reviewer's comments. These results should be included in this manuscript. The contribution of AMPK (line 261) is still unclear.

Response: Thanks for your question. We have added the results that AC9 is the target gene of miR-143 in the main text (Figure 5). We deleted the relevant description of AMPK and added the possible signal pathway of AC9 regulating UCP1 in the discussion section (line 298-300).

  1. The regulation of miR-143 by IL6 should be discussed in the section of discussion.

Response: Thanks for your suggestion. The regulation of miR-143 by IL6 has been discussed in the section of discussion (line 271-275).

  1. Statistical analysis: the meaning of a, b and c is unclear (Figure 2F-H, Figure 3G-I, and Figure 5).

Response: Thanks for your question. The meanings of “a”, “b”, and “c” have been defined in the figure legends (different letters above groups represent significant differences, with shared letters representing no significant differences, as determined by two-way ANOVA followed by Tukey’s post hoc test).

Reviewer 2 Report

The author has addressed some of my concerns in the revised manuscript. However, to get such a conclusion, more experiments are still needed as commented before. Also, some results in the response are contradictory and inconsistent with the major conclusion of this manuscript. 

  1. Fig.1G and 1H “miR-143 directly target AC9” in the author’s response indicate that UCP1 expression is significantly increased in miR-143 KO mice which is contradictory with the result shown in Fig.2A “Thermogenic-related genes and inflammation-related genes expression in BAT of mice fed normal diet” that showed comparable UCP1 expression between WT and miR-143 KO mice. 
  2. As commented before, in Figures 5A and 5B, LPS treatment can induce AC9 and UCP1 levels in miR-143 KO brown adipocyte, indicating that miR-143 is not essential for LPS-induced thermogenesis, which is inconsistent with the major conclusion in this manuscript that LPS can directly regulate adipocyte thermogenesis through miR-143/ adenylate cyclase. 
  3. As SVF cells could express miR-143, it would be better to use the freshly isolated mature adipocytes and SVF cells to rule out the possibility that SVF cells contribute to the LPS-induced miR-143 expression in BAT. 

Author Response

The author has addressed some of my concerns in the revised manuscript. However, to get such a conclusion, more experiments are still needed as commented before. Also, some results in the response are contradictory and inconsistent with the major conclusion of this manuscript.

  1. Fig.1G and 1H “miR-143 directly target AC9” in the author’s response indicate that UCP1 expression is significantly increased in miR-143 KO mice which is contradictory with the result shown in Fig.2A “Thermogenic-related genes and inflammation-related genes expression in BAT of mice fed normal diet” that showed comparable UCP1 expression between WT and miR-143 KO mice.

Response: Thanks for your question. Fig.1G and 1H “miR-143 directly target AC9” (as in the main text Fig. 5I and 5J) indicate that UCP1 expression is significantly increased in miR-143KO primary brown adipocytes (not in mice). The expression of UCP1 between WT and miR-143KO mice and WT and miR-143KO primary brown adipocytes was indeed inconsistent. Many studies have found that the results of in vivo and in vitro experiments are inconsistent, such as leptin, IL-9 and IL-10 (1-3), which may be caused by differences between the cellular environment and the conditions in vitro.

  1. As commented before, in Figures 5A and 5B, LPS treatment can induce AC9 and UCP1 levels in miR-143 KO brown adipocyte, indicating that miR-143 is not essential for LPS-induced thermogenesis, which is inconsistent with the major conclusion in this manuscript that LPS can directly regulate adipocyte thermogenesis through miR-143/ adenylate cyclase.

Response: Thanks for your question. We agree with the idea that miR-143 is not essential for LPS-induced thermogenesis, but miR-143 do participate in the regulation of thermogenesis induced by LPS. It has been reported that miR-143 is highly expressed in BAT (4,5), indicating its significance in BAT. Our study found that in brown adipocytes, miR-143 inhibited thermogenesis by down-regulating UCP1 (main text Fig. 3G–I), and LPS inhibited miR-143 expression in order to relieve the inhibition of miR-143 on thermogenesis. miR-143KO diminished negative regulation of UCP1, thus LPS stimulation further increased the thermogenesis of miR-143KO brown adipocytes (WT+LPS group VS. miR-143KO+LPS group). Thus, we revised our conclusion as LPS increased BAT thermogenesis by a way of inhibiting miR-143 expression, a negative regulator for AC9.

  1. As SVF cells could express miR-143, it would be better to use the freshly isolated mature adipocytes and SVF cells to rule out the possibility that SVF cells contribute to the LPS-induced miR-143 expression in BAT.

Response: According to your suggestion, we isolated SVF cells and mature adipocytes from BAT of mice treated with or without LPS. The results showed that LPS significantly reduced the expression of miR-143 in mature adipocytes, but did not affect the expression of miR-143 in SVF cells (main text Fig. 3A).

  1. Soroosh P, Doherty TA. Th9 and allergic disease. Immunology. 2009;127(4):450-8.
  2. Endo H, Hosono K, Uchiyama T, Sakai E, Sugiyama M, Takahashi H, et al. Leptin acts as a growth factor for colorectal tumours at stages subsequent to tumour initiation in murine colon carcinogenesis. Gut. 2011;60(10):1363-71.
  3. Huibregtse IL, van Lent AU, van Deventer SJ. Immunopathogenesis of IBD: insufficient suppressor function in the gut? Gut. 2007;56(4):584-92.
  4. Elia, L.; Quintavalle, M.; Zhang, J.; Contu, R.; Cossu, L.; Latronico, M.V.; Peterson, K.L.; Indolfi, C.; Catalucci, D.; Chen, J.; et al. The knockout of miR-143 and -145 alters smooth muscle cell maintenance and vascular homeostasis in mice: correlates with human disease. Cell Death Differ 2009, 16, 1590–1598, doi:10.1038/cdd.2009.153.
  5. Jordan, S.D.; Kruger, M.; Willmes, D.M.; Redemann, N.; Wunderlich, F.T.; Bronneke, H.S.; Merkwirth, C.; Kashkar, H.; Olkkonen, V.M.; Bottger, T.; et al. Obesity-induced overexpression of miRNA-143 inhibits insulin-stimulated AKT activation and impairs glucose metabolism. Nat Cell Biol 2011, 13, 434-446, doi:10.1038/ncb2211.
